# GENERATIVE MODELING FOR PROTEIN STRUCTUES

**Namrata Anand & Possu Huang**
Bioengineering Department
Stanford University
{namrataa, possu}@stanford.edu

## ABSTRACT

We apply deep generative models to the task of generating protein structures, toward application in protein design. We encode protein structures in terms of pairwise distances between alpha-carbons on the protein backbone, which by construction eliminates the need for the generative model to learn translational and rotational symmetries. We then introduce a convex formulation of corruption-robust 3-D structure recovery to fold protein structures from generated pairwise distance matrices, and solve this optimization problem using the Alternating Direction Method of Multipliers. Finally, we demonstrate the effectiveness of our models by predicting completions of corrupted protein structures and show that in many cases the models infer biochemically viable solutions.

## 1 INTRODUCTION

In this paper, we use Generative Adversarial Networks (GANs) to generate novel protein structures (Goodfellow et al. (2014); Radford et al. (2015)) and use our trained models to predict missing sections of corrupted protein structures. Analyzing the form and function of proteins is a key part of understanding biology at the molecular and cellular level. In addition, a major engineering challenge is to design new proteins in a principled and methodical way. So far, progress in computational protein design has led to the development of new therapies (Whitehead et al. (2012); Strauch et al. (2017)), enzymes (Röthlisberger et al. (2008); Siegel et al. (2010); Jiang et al. (2008)), small-molecule binders (Tinberg et al. (2013)), and biosensors (Smart et al. (2017)). These efforts are largely limited to modifying naturally occurring, or "native," proteins. To fully control the structure and function of engineered proteins, it is ideal in practice to design proteins *de novo* (Huang et al. (2016)).

Proteins are macromolecules made up of chains of amino acids. Each amino acid has an amine group, a carboxyl group, and one of 20 side chain ("R") groups. The protein backbone formed has a repeating pattern of amine and carbonyl groups, with various side chains branching off the backbone. Interactions between side chains and the protein backbone give rise to local secondary structure elements– such as rigid helices and sheets, or more flexible loops– and to the ultimate tertiary structure of the protein.

We use a data representation restricted to structural information– pairwise distances of alpha-carbons on the protein backbone. Despite this reduced representation, our method successfully learns to generate new structures and, importantly, in many cases infers viable solutions for completing corrupted structures. We use the Alternating Direction Method of Multipliers (ADMM) algorithm (Boyd et al. (2011)) to fold 2-D pairwise distance "maps" into 3-D Cartesian coordinates in order to evaluate the generated structures.

The protein design process can be roughly separated into two parts: First, designing the scaffold of a structure, and second, finding the sequence of amino acids or residues which will fold into that structure. In practice, this is done by optimizing an energy function determined from native structures. This is the operation at the heart of Rosetta– a widely-used tool in computational protein design (Huang et al. (2016); Leaver-Fay et al. (2011); Huang et al. (2011)). Rosetta samples native protein fragments to fold backbones and then optimizes over amino acid types and orientations ("rotamers") to determine the most likely amino acid sequence of the designed protein (Das & Baker (2008)). We focus on the first part of this design process – generating and evaluating new

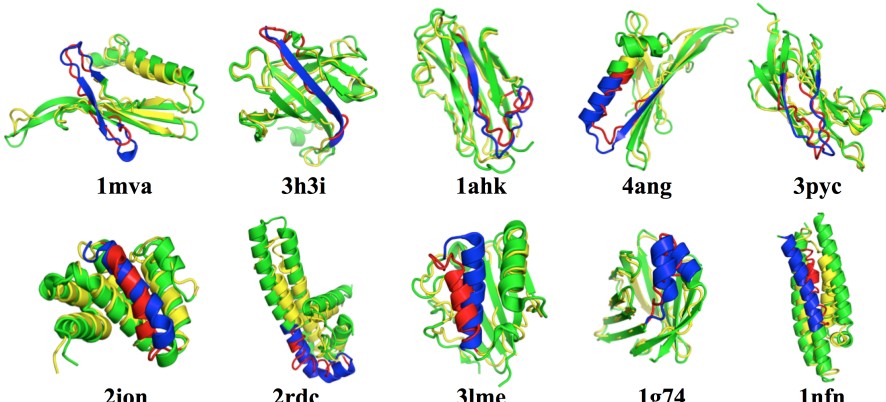

**1mva**     **3h3i**     **1ahk**     **4ang**     **3pyc**

**2ion**     **2rdc**     **3lme**     **1g74**     **1nfn**

Figure 1: Examples of 20-residue inpainting solutions for 128-residue structures folded using ADMM (PDB ID listed under structure). Native structures are colored green and reconstructed structures are colored yellow. The omitted regions of each native structure are colored blue, and the inpainted solutions are colored red.

protein structures. We use Rosetta as a baseline for speed and accuracy, folding structures directly from the generated pairwise distance constraints between alpha-carbons.

The main contributions of this paper are (i) a generative model of proteins that estimates the distribution of their structures in a way that is invariant to rotational and translational symmetry and (ii) a differentiable, corruption-robust convex formulation for the resulting reconstruction problem that scales to large problem instances.

## 2 METHODS AND RESULTS

**Generating maps.** We use deep convolutional generative adversarial networks (DCGANs) as our generative models for protein structure (Radford et al. (2015)). Our data source is the Protein Data Bank (Berman et al. (2000)), a repository of experimentally determined structures available on-line. We encode 3-D structure as 2-D pairwise distances ("maps") between alpha-carbons on the protein backbone. Note that the maps preserve the order of the peptide chain from N- to C- terminus.

We generated 16-, 64-, 128-, and 256-residue maps by training GANs on non-overlapping fragments of the same lengths from PDB structures. Importantly, the inputs to the model are not all distinct domains which fold *in vivo*, but are fragments. Therefore, the model is not necessarily learning protein structures which will fold, but rather is learning realistic secondary and tertiary structural elements.

**Folding maps via ADMM.** In practice, using Rosetta's optimization toolkit to find a low-energy structure via sampling takes on the order of tens of minutes to fold small structures of less than 100 residues because of the fragment and rotamer sampling steps. Therefore we sought another, faster way to reconstruct 3-D protein structure via the ADMM algorithm (Boyd et al. (2011)), which is a combination of dual decomposition and the method of multipliers.

The task of determining 3-D cartesian coordinates given pairwise distance measurements is already well-understood and has a natural formulation as a convex problem (Boyd & Vandenberghe (2004)). Given $m$ coordinates $[a_1, a_2, \ldots a_m] = A \in \mathbb{R}^{n \times m}$, the Gram matrix $G = A^T A \in \mathcal{S}_+^m$. Note that $G$ is symmetric, positive-semidefinite with rank at most $n$. We want to recover $A$ given pairwise distance measurements $D$, with $d_{ij} = \|a_i - a_j\|_2$. Since $g_{ij} = a_i^T a_j$ and $d_{ij}^2 = g_{ii} + g_{jj} - 2g_{ij}$, we can find $G$ by solving an SDP over the positive semidefinite cone. While this optimization problem can be solved quickly using SDP solvers for systems where $n$ is small, the runtime of traditional solvers is quadratic in $n$ and renders large structure recovery problems out of reach. Hence, we use

ADMM which we found converges to the correct solution quickly. We write a modified optimization problem

$$\min_{G,Z,\eta} \ \lambda \, \|\eta\|_1 + \frac{1}{2} \left( \sum_{i=1,j=1}^{m} (g_{ii} + g_{jj} - 2g_{ij} + \eta_{ij} - d_{ij}^2)^2 \right) + \mathbb{1}\{Z \in \mathcal{S}_+^m\} \tag{1}$$

$$\text{subject to } G - Z = 0$$

where we have allowed a slack term $\eta$ on each distance measurement, whose $\ell_1$ norm is penalized. Now we can decompose the problem into iterative updates over variables $G$, $Z$, and $U$ as

$$
\begin{aligned}
G_{k+1}, \eta_{k+1} &= \underset{G,\eta}{\operatorname{argmin}} \ \left[ \, \lambda \, \|\eta\|_1 + \frac{1}{2} \left( \sum_{i=1,j=1}^{m} (g_{ii} + g_{jj} - 2g_{ij} + \eta_{ij} - d_{ij}^2)^2 \right) + \frac{\rho}{2} \, \|G - Z_k + U_k\|_2^2 \, \right] \\
Z_{k+1} &= \Pi_{\mathcal{S}_+^m}(G_{k+1} + U_k) \\
U_{k+1} &= U_k + G_{k+1} - Z_{k+1}
\end{aligned}
\tag{2}
$$

with augmented Lagrangian penalty $\rho > 0$. The update for $Z$ is simply the projection onto the set of symmetric positive semidefinite matrices of rank $n$. We find $G_k$ and $\eta_k$ by several iterations of gradient descent. After convergence, coordinates $A$ can be recovered from $G$ via SVD. Note that this algorithm is generally applicable to any problem for structure recovery from pairwise distance measurements, not only for protein structures.

**Inpainting for protein design.** We considered how to use the trained generative models to infer contextually correct missing portions of protein structures. We can formulate this problem as an inpainting problem, where for a subset of residues all pairwise distances are eliminated and the task is to design a new segment reasonably, given the context of the rest of the uncorrupted structure.

We used a modified version of the semantic inpainting method described in Yeh et al. (2016), omitting the Poisson blending step. To minimize structural homology between the GAN training data and the inpainting test data, we separated train and test structures by classified fold type (Murzin et al. (1995)). Results for inpainting of missing residues on 128-residue maps are shown in Figure 2. It is clear that the trained GAN can fill in semantically correct pairwise distances for the removed polypeptide sequences.

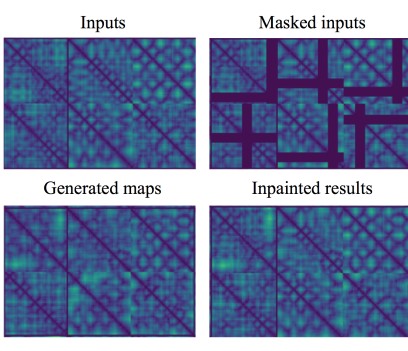

Inputs     Masked inputs

Generated maps     Inpainted results

Figure 2: Examples of inpainting for 20 missing residues on 128-residue maps.

To test whether these inpainted portions correspond to legitimate reconstructions of missing parts of the protein, we used ADMM to fold the new inpainted maps into structures. Some solutions found are shown in Figure 1.

## 3 CONCLUSION

We use GANs to generate protein alpha-carbon pairwise distance maps and use ADMM to "fold" the protein structure. We apply this method to the task of inferring completions for missing residues in protein structures. We can extend our generative modeling procedure to solve the structure recovery problem end-to-end. The current approach factors through the map representation, which overconstrains the recovery problem. By incorporating the ADMM recovery procedure as a differentiable optimization layer of the generator, we can extend the models presented in this paper to directly generate and evaluate 3-D structures.

### ACKNOWLEDGMENTS

We would like to thank Frank DiMaio for helpful discussion on Rosetta and providing baseline scripts to fold structures directly from pairiwse distances.

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

## A GAN TRAINING DETAILS

For all our models we used a fixed noise vector size of 100 units. Many of our models show inherent instability, but usually only after converging to a very good solution for map generation. While we implemented various methods for stabilizing GAN training (Metz et al. (2016); Arjovsky et al. (2017); Gulrajani et al. (2017); Salimans et al. (2016)), we found that in practice for this problem, these were not necessary for training a good model.

For upsampling by the generator, we use strided convolution transpose operations instead of pixel shuffling (Shi et al. (2016)) or interpolation, as we found this to work better in practice. We typically set the slope of the LeakyReLU units to 0.2 and the dropout rate to 0.1 during training. We did not normalize input maps but scaled them down by a constant factor. During training, we enforce that $G(\mathbf{z})$ be symmetric by setting $G(\mathbf{z}) \leftarrow \frac{G(\mathbf{z}) + G(\mathbf{z})^T}{2}$ before passing the generated map to the discriminator. We train our models using the Adam optimizer (Kingma & Ba (2014)). All models were implemented using PyTorch (Paszke et al. (2017)).

Folded structures from our trained 64-residue model are shown in Figure 3, alongside examples from the training set. We found that the generator was able to learn to construct meaningful secondary structure elements such as alpha helices and beta sheets, with a bias toward the former.

Real samples            Generated samples

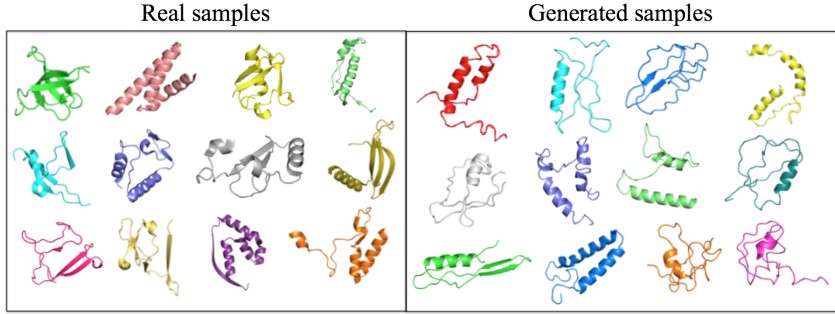

Figure 3: Examples of real (left) 64-residue fragments from the training dataset versus generated (right) 64-residue fragments folded subject to distance constraints using Rosetta.

### A.1 TESTING THE COMPLEXITY OF THE GAN

We asked whether for all native structures $\mathbf{x}$ we could find a corresponding $\mathbf{z} \in \mathbb{R}^n$ such that $G(\mathbf{z}) \approx \mathbf{x}$. To do this, we optimized $\mathbf{z}$ using pretrained GANs with a modified reconstruction loss objective $L_z$, adding a K-L divergence regularizer term $L_{KL}$ over the mean and variance of elements of $\mathbf{z}$.

$$L_z(\{\mathbf{z_1}, \mathbf{z_2} \ldots, \mathbf{z_m}\}) = \frac{1}{m} \sum_{j=1}^{m} (\|G(\mathbf{z_j}) - \mathbf{x_j}\|_2) + \gamma \, L_{KL}(\{\mathbf{z_1}, \mathbf{z_2} \ldots, \mathbf{z_m}\}) \tag{3}$$

$$L_{KL}(\{\mathbf{z_1}, \mathbf{z_2} \ldots, \mathbf{z_m}\}) = -\frac{1}{2} \sum_{i=1}^{n} (1 + \log(\sigma_i^2) - \mu_i^2 - \sigma_i^2) \tag{4}$$

In practice, we calculate $\mu_i, \sigma_i$ over a large batch of vectors and set $\gamma = 10$. Results for recovery of 64-residue and 128-residue structures are shown in Figure 4. We successfully recover maps with most of the input structural details. For the 128-residue maps, occasionally details are lost in the recovered map, which suggests perhaps we need to increase the complexity of that model.

Real maps       Recovered maps

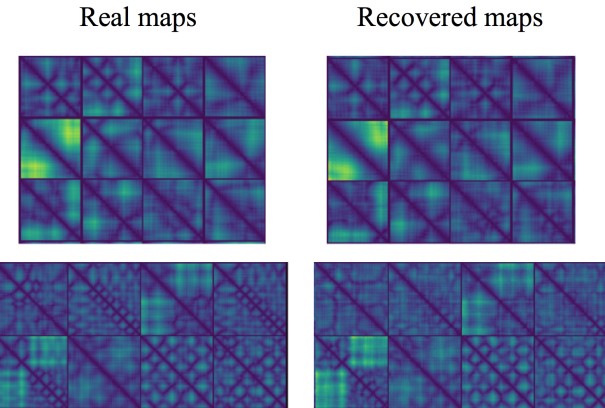

Figure 4: Recovery of maps for 64-residue (top) and 128-residue (bottom) models by optimization of GAN input vector **z**

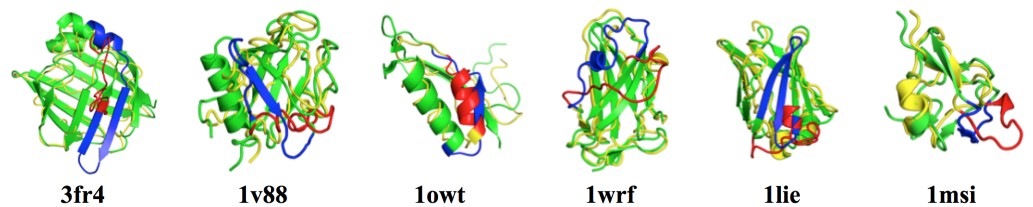

Figure 5: Examples of non-native and incorrect 20-residue (left) and 10-residue (right) inpainting solutions for selected 128-residue and 64-residue structures, respectively, folded using ADMM (PDB ID listed under structure). Native structures are colored green and reconstructed structures are colored yellow. The omitted regions of each native structure are colored blue, and the inpainted solutions are colored red.

## B  CORRUPTION ROBUSTNESS OF FOLDING

Our folding procedure via ADMM is fairly robust to corruption of data. In Figure 6, we show folding error of structures versus log fraction of corruptions $\log m$ in pairwise distances with Lagrangian penalty weight $\rho = 10$ and varying slack weight $\lambda$ and noise $\sim$Unif$[-c, c]$. The error is calculated by doing least-squares rigid-body alignment of the new coordinates with respect to the coordinates for the true structure. We see that for $c = 5$ and $c = 10$, the rigid-body alignment error is roughly constant until about 10% of the pairwise distance measurements are corrupted.

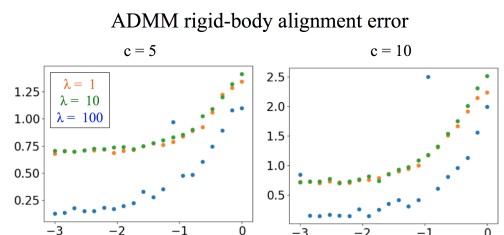

Figure 6: Mean rigid-body alignment error of structures versus log fraction of corruptions $\log m$

## C  INPAINTING EXPERIMENT DETAILS AND ANALYSIS

For inpainting, we optimize the input vector **z** of the GAN to find a fake image which, when overlayed over the masked region of the input, gives a good inpainting solution. There are three loss terms optimized for this procedure. The first is a context loss term, which is an $\ell_1$ reconstruction

loss with higher weighting for pixels nearer to the masked region of the input. Given input $\mathbf{x}$ and binary mask $M$ delineating the area to be inpainted, the weighting term $W$ is found by convolving the mask complement $M^C$ with a 2-D identity filter of fixed size. For our experiments with 64-residue and 128-residue maps, we set the filter sizes to $9 \times 9$ and $15 \times 15$, respectively. The context loss is

$$L_{\text{context}}(\mathbf{z}) = \|(W * M^C) * (G(\mathbf{z}) - \mathbf{x})\|_1 \tag{5}$$

The next loss term is a prior discriminator loss with respect to the generated image used for the inpainting.

$$L_{\text{prior}}(\mathbf{z}) = \log(1 - D(G(\mathbf{z}))) \tag{6}$$

Finally, there is the discriminator loss on the final inpainting solution.

$$L_{\text{disc}}(\mathbf{z}) = \log(1 - D(M * G(\mathbf{z}) + M^C * \mathbf{x})) \tag{7}$$

The full objective is

$$\min_{\mathbf{z}} \; L_{\text{context}}(\mathbf{z}) + \gamma \, L_{\text{prior}}(\mathbf{z}) + L_{\text{disc}}(\mathbf{z}) \tag{8}$$

where we set weighting term $\gamma = 0.003$. As during training, we enforce that the generator output $G(\mathbf{z})$ be symmetric.

In Figure 7 we see that as the inpainting task becomes harder, the discriminator score for the inpainting solution decreases and the rigid-body alignment error increases. However, in the case when the inpainted solution is good but deviates from the native structure, the rigid-body alignment error will be high. Therefore, we cannot necessarily view this metric as a good indicator of whether the reconstructed solution is reasonable.

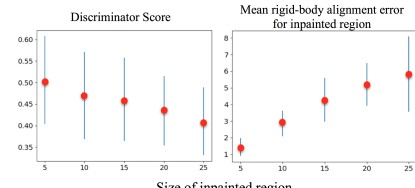

Figure 7: Discriminator score and mean coordinate $\ell_2$ alignment error for 64-residue inpainting task

In Figure 5, we show some non-native and incorrect inpainting solutions found. The model sometimes renders a solution that does not align with the native structure, as in *3fr4* where the solution is to lengthen a loop between a helix and beta strand, or in *1owt* where a short beta strand is replaced by a helix of similar length and position. In some cases, the model makes obvious mistakes, for example, in *1v88* where the a beta sheet is replaced with a helix-like loop that folds in unnaturally on itself.

