# OpenReview forum: "Generative Modeling for Protein Structures"
_ICLR.cc/2018/Workshop — Accept_

### Official Review · AnonReviewer1 · 2018-03-07
**Interesting application of GAN to protein structure discovery, but some more model details needed**

**Rating:** 6
**Confidence:** 4

**Review:**

- This is a very interesting new application of GAN, where the authors train a generative model on short protein structural fragments. This leads to interesting application of doing 'inpainting' of protein structures when there are missing residues, or when certain fragments need to be re-designed.

- Another contribution of this paper is the use of ADMM to recover the protein structure from pairwise distance matrices, instead of the more expensive option of running an energy optmization tool like Rosetta.

- I would like to see more details on the GAN model itself. Appendix C shows the loss functions used in the inpainting experiments, but no architecture details on the neural networks.

- It will be good to give some more quantitative analysis on the performance of this methods. For example, what's the reconstruction error of the method on a sample of PDB structures of reasonable size? It would also be interesting to look at the error distribution on different secondary structures, such as whether the loop regions have larger errors compared to helices and beta sheets.

- Overall I think this paper is a very interesting application of GAN in the computational biology domain, but I would like to see more model details and error analysis for a machine learning audience.

---

### Official Review · AnonReviewer3 · 2018-03-09
**Clearly written and well motivated. However, the experiments are mainly qualitative in nature. Thus, the significance of this work for the protein design field is a bit unclear to me.**

**Rating:** 6
**Confidence:** 3

**Review:**

SUMMARY
This work is on protein design, i.e.~generating protein structures. Based of a large repository of proteins, in a first step a convolutional generative GAN is trained on (length-matched) fragments in order to produce 2D pairwise distance maps between alpha carbons on the backbone. In a second step, protein folding maps are computed by the ADMM algorithm. In particular, the problem of recovering 3D coordinates from pairwise distances is addressed by iteratively projecting an updated gram matrix onto the cone of symmetric positive semidefinite matrices of rank 3.  Finally, the proposed method is applied to the problem of "inpainting" missing parts of proteins.

EVALUATION:
The paper is easy to read, the motivation is clear and convincing, and everything seems to be technically sound. The proposed technique for recovering 3D coordinates from pairwise distances is certainly interesting, but it is a bit unclear to me if there is any specific novel contribution. The "inpainting" application example is certainly interesting on a qualitative level. Unfortunately, there is (almost) no quantitative evaluation provided, which makes it a bit difficult for the reader to understand what type of problems can be solved successfully, and where the limits of this approach are.

---

### Official Review · AnonReviewer2 · 2018-03-11
**Good application of ADMM / GAN for contact reconstruction**

**Rating:** 8
**Confidence:** 3

**Review:**

Original application and seem to be significant. Clear and detailed delivery of the work.

---

### Public Comment · ~Oriol_Vinyals1 · 2018-02-17
**Please Fix Length**

Your paper violates by a few lines the 3 page limit (see https://iclr.cc/Conferences/2018/CallForWorkshops). Please send us a fixed version of your PDF at iclr2018.programchairs@gmail.com by the end of Monday, February 19th, or else we will reject your paper.

Thanks,
ICLR2018 Program Chairs

---

### Decision · Program_Chairs · 2018-03-20
**ICLR 2018 Workshop Acceptance Decision**

**Decision:**

Accept

**Comment:**

Congratulations, your paper was accepted to the ICLR workshop.